# Accessibility of Psychological Treatments for Bulimia Nervosa: A Review of Efficacy and Engagement in Online Self-Help Treatments

**DOI:** 10.3390/ijerph20010119

**Published:** 2022-12-22

**Authors:** Sarah Barakat, Sarah Maguire

**Affiliations:** 1InsideOut Institute for Eating Disorders, University of Sydney, Sydney Local Health District, Camperdown 2050, Australia; 2School of Psychology, University of Sydney, Camperdown 2050, Australia

**Keywords:** eating disorders, bulimia nervosa, psychological treatment, cognitive behaviour therapy, self-help treatment, online treatments

## Abstract

Bulimia nervosa is an eating disorder characterised by marked impairment to one’s physical health and social functioning, as well as high rates of chronicity and comorbidity. This literature review aims to summarise existing academic research related to the symptom profile of BN, the costs and burden imposed by the illness, barriers to the receipt of care, and the evidence base for available psychological treatments. As a consequence of well-documented difficulties in accessing evidence-based treatments for eating disorders, efforts have been made towards developing innovative, diverse channels to deliver treatment, with several of these attempting to harness the potential of digital platforms. In response to the increasing number of trials investigating the utility of online treatments, this paper provides a critical review of previous attempts to examine digital interventions in the treatment of eating disorders. The results of a focused literature review are presented, including a detailed synthesis of a knowledgeable selection of high-quality articles with the aim of providing an update on the current state of research in the field. The results of the review highlight the potential for online self-help treatments to produce moderately sized reductions in core behavioural and cognitive symptoms of eating disorders. However, concern is raised regarding the methodological limitations of previous research in the field, as well as the high rates of dropout and poor adherence reported across most studies. The review suggests directions for future research, including the need to replicate previous findings using rigorous study design and methodology, as well as further investigation regarding the utility of clinician support and interactive digital features as potential mechanisms for offsetting low rates of engagement with online treatments.

## 1. Introduction

### 1.1. Bulimia Nervosa

Bulimia nervosa (BN) is an eating disorder (ED) characterised by recurrent binge-eating episodes and inappropriate compensatory behaviours, organised around marked concern with weight and shape [1]. BN was first formally described as a variant of anorexia nervosa (AN) in 1979 by psychiatrist Gerald Russell [2]. Prior to this, early precursors of the term “bulimia” were noted in the ancient Greek literature with the origin word *bulimy* translating to “ravenous hunger” [3]. Ancient Roman aristocracy were also observed to engage in gorging and vomiting behaviour, yet lack of reference to concerns regarding shape or weight for such individuals suggest that these presentations were likely distinct from modern conceptualisations of BN [4]. Descriptions of bulimic episodes as being motivated by weight-related concerns first emerged in psychiatric case reports as early as 1903; however, clinical presentations of bulimic symptoms remained fairly absent until the 1960s to 1970s, at which point clinicians began to note an increasing number of cases [5]. This increase in prevalence was conceptualised as a response to changes in the sociocultural environment, in particular to a growing emphasis upon the thin ideal [6]. Following Russell’s (1979) paper, further research was devoted to BN, culminating in the distinction of bulimia in the third edition of the Diagnostic and Statistical Manual of Mental Disorders (DSM-3) [7] as an ED independent of anorexia nervosa (AN) and deserving of clinical attention in its own right [3].

Since such early descriptions and conceptualisations of BN, the defining features and clinical profile of the disorder have evolved. The following literature review aims to provide a summary of the evidence to date regarding the epidemiology and aetiology of BN, as well as the evolution of effective and accessible treatments. The methodology adopted as part of the current review is consistent with that of a focused literature review [8]. Accordingly, we aim to present evidence from a knowledgeable selection of relevant, high-quality articles, including both individual trials of psychological interventions for bulimia nervosa as well as previous systematic reviews and meta-analyses in the field. Given that the review aims to provide a broad-reaching overview of the current state of research on self-help interventions for bulimia nervosa, no eligibility criteria were applied for included studies.

#### 1.1.1. Clinical Description of Bulimia Nervosa

In 1979, Russell proposed two criteria to diagnose BN: (1) an irresistible urge to overeat, followed by self-induced vomiting or purging and (2) a morbid fear of becoming fat [2]. Since then, more comprehensive diagnostic criteria have been developed which incorporate growing knowledge of the symptoms associated with BN. The most recent iteration of the diagnostic criteria for BN according to DSM, Version 5, Text Revision (DSM-5-TR), are listed in Table 1 [1]. The definition of episodes of overeating has been refined to specify that binge-eating episodes must involve the consumption of an objectively large amount of food within a discrete period of time, usually two hours. An additional key criterion is that the individual must experience a loss of control during the binge episode, often experienced by patients as feeling “numb” or “nothing”. Objective binge episodes are distinguished from subjective binge episodes, in which there is a similar experience of loss of control, however the amount of food consumed is not considered to be larger than what most individuals would eat in the given context [9]. Binge episodes usually consist of foods that are easy to ingest and later purged or those which the individual avoids due to fears of weight gain, commonly high-calorie foods [3]. In terms of caloric intake, objective binge episodes often range from 4200 to 8400 kJ in size [10].

A further advancement upon Russell’s early criteria includes a broadening of the range of compensatory behaviours beyond self-induced vomiting. Individuals affected by BN may also engage in excessive exercise, fasting, extreme dietary restriction, or misuse of laxatives, diuretics, or other medications [1]. Accordingly, BN can be characterised into two subtypes which differ according to the type of compensatory behaviour endorsed. The BN purging subtype includes individuals who primarily engage in self-induced purging and the BN non-purging subtype refers to those who regularly engage in other behaviours designed to further reduce weight [1]. On average, 76% of women diagnosed with BN are classified within the purging subtype and 24% are classified in the non-purging subtype [11]. Irrespective of type, all compensatory behaviours are understood as being driven by a fear of weight gain and serve to further reinforce the core psychopathology of an overvaluation of shape and weight [12,13].

Individuals affected by BN commonly possess a body weight considered to be average or slightly above average for their age and height [14]. However, one’s weight may fluctuate marginally depending upon the individual’s pattern of binge eating, restriction, and other compensatory behaviours. Symptoms of BN can be difficult for friends and family to detect as the behaviours of binge eating and compensation are often hidden. Such secrecy is indicative of the pervasive shame and embarrassment surrounding the disordered eating and represents a key psychological feature of the illness [15].

#### 1.1.2. Epidemiology

A noteworthy study examining the prevalence of BN and other EDs in the Australian population was recently conducted by Bagaric and colleagues (2020). Using a sample of 2977 Australians aged 15 years and over, Bagaric et al. (2020) found the estimated lifetime prevalence rate of BN to be 3.8%. On average, the lifetime prevalence rate for females was twice as high than that for males, with estimates of 2.59% and 1.21%, respectively [16]. The study also estimated the average point prevalence of BN in a female population to be between 0.7% and 0.81%. Studies of other Western populations and broader reviews of the literature report lifetime prevalence rates of up to 3% in females and more than 1% in males [17,18,19,20]. The prevalence of BN has been argued to be much greater than the number of cases which meet the stringent DSM-5 criteria [21]. Rather, it is estimated that the prevalence rate of individuals who fall within the subclinical category of BN, known as other specified feeding or eating disorder (OSFED) with bulimic behaviours, could be as high as 14% to 22% [22,23].

Amongst the wider literature, the highest BN prevalence rates have been consistently reported in younger age groups [24,25]. This has been confirmed by van Eeden, Hoeken, and Hoek’s (2021) updated review of global prevalence studies which reported the peak age of incidence of BN to be between 15 and 29 years [20].

Research into illness onset has found that BN typically begins during young adulthood, most commonly between 16 and 20 years of age, with binge-eating episodes identified as one of the first symptoms to appear [18,26]. This differs from AN which has a slightly younger age of symptom onset of 12 to 14 years with early signs commonly involving dietary restriction [27]. Additionally, there is evidence to suggest the age of onset of BN has decreased overtime. Favaro et al. (2009) analysed time trends in illness onset in a sample of 793 patients with BN referred to an outpatient ED unit between 1985 and 2008. The authors found the mean age of BN onset in the whole sample to be 19.3 years, with 65% of patients affected before 20 years of age. Further analyses revealed a significant decrease in the age of onset from 18.5 years in the period of 1970–1972 to 17.1 years in the period of 1979–1981 [28].

#### 1.1.3. Burden and Cost

The impairment caused by BN is widespread, affecting one’s social, psychological and occupational functioning in such a way that its impact can persist long beyond the disorder itself [29]. The complexity of BN is compounded by high rates of comorbidity with chronic physical health conditions as well as other psychiatric disorders [26]. Medical complications associated with BN are often related to the method and frequency of compensatory behaviours, in particular self-induced vomiting and laxative use. These include dental enamel erosion, oesophageal damage, gastrointestinal dysfunction, and electrolyte imbalance, which can lead to cardiac malfunction, hospitalisation, and even death [30,31]. Individuals with BN have a weighted mortality rate of 1.74 per 1000 person-years, which is estimated to be two times higher than their age- and sex-matched peers in the general population [20,32].

Of all EDs, BN upholds the highest rate of comorbidity with other psychiatric disorders. It is estimated that between 84% and 94% of individuals with BN will meet lifetime criteria for another mental health disorder, including anxiety, mood, impulse-control, and substance-use disorders [24,26]. Major depressive disorder has been identified as the most commonly diagnosed comorbid disorder alongside BN, with data suggesting this cohort has more severe psychopathology and poorer treatment outcomes [33,34]. Additionally, there is evidence to suggest a strong association between BN and specific Axis II disorders, in particular borderline personality disorder (BPD, [35]). Research has identified commonalities between the core problematic behaviours evident in BN (binge eating and purging) and BPD (self-harm, suicide attempts, substance abuse), such as that they are both associated with elevated levels of impulsivity and a poor capacity for emotion regulation [36].

The financial costs of BN are substantial. Expenses incurred by the individual include those associated with the symptoms of the illness (e.g., purchase of food consumed during a binge episode) as well as the cost of treatment and general management of symptoms caused by the illness [37,38]. Crow et al. (2009) assessed the costs associated with binge eating in a sample of participants with BN. They found 32.7% of one’s total annual food costs were associated with binge eating and purging, summing to USD 1599.45 per year on average [39]. Additionally, individuals living with an ED are subject to yearly healthcare costs that are 48% higher than the general population’s [40].

The cost of BN to the economy is twofold including: (1) indirect loss of productivity via absence from the education system or workforce and (2) direct costs of healthcare access [41,42]. Streatfeild et al. (2021) estimated the total costs associated with EDs to be USD 64.7 billion, with BN accounting for 18% of these total costs [43]. Within the Australian healthcare system, a nationwide report of the economic burden of EDs estimated the cost to be AUD 7.6 million in the year 2008–2009 [44]. More recently, Tannous et al. (2021) conducted a health economics analysis of a community-based study of 2977 participants living in South Australia. The authors calculated the annual total economic cost of EDs in 2018 to be AUD 84 billion for South Australia alone, with the majority of costs (AUD 81 billion) accounted for by the burden of the disease (i.e., years lived with disability and years of life lost) and a comparatively smaller amount accounted for by health-system costs (AUD 1 billion) [45]. This is consistent with research regarding elevated functional impairment associated with the illness. For example, Mond and Hay (2007) found that in a sample of 1757 women, just over one third of those endorsing bulimic behaviours reported absence from their employment for at least one day in the preceding four weeks [46].

#### 1.1.4. Course of Illness

BN often runs a chronic course [47,48]. Repeated treatment failures are not uncommon and it is estimated that half of patients with BN will not recover in response to an adequate treatment course [49]. As part of a large multinational community-based study conducted by Kessler et al. (2013) (*n* = 24,124), the persistence of BN was found to be higher than BED, with median years in episode reported to be 6.5 (range: 2.2–15.4) for BN and 4.3 (range: 1.0–11.7) for BED [26]. Similarly, a longitudinal community study of 3021 German participants found that 42% of individuals who met DSM-4 criteria for BN at baseline continued to endorse ED symptomology at any of the three follow-up assessments conducted at 1.6 years, 3.5 years, and 8.2 years post initial assessment [18].

A longitudinal study by Eddy et al. (2017) was one of the first to assess the course of EDs beyond 20 years of follow-up. The study recruited a sample of 246 participants, including 136 diagnosed with AN and 110 with BN, and assessed them at 9- and 22-year follow-ups. It was found that 68.2% of BN patients recovered by the 9-year follow-up as compared to 31.4% of AN patients, whereas at 22-year follow-up, the recovery rates remained stable for BN at 68.2% yet the proportion of recovered AN cases increased to 62.8%. The authors conclude that although recovery from BN may occur more rapidly, the illness is more likely than AN to persist if early change is not achieved [50]. Very few predictors have been consistently associated with better prognosis; however, those which have shown some association include shorter illness duration, less severe bulimic symptomology, and younger age of treatment seeking [3].

## 2. Treatment Barriers

The chronicity and complexity of BN are compounded by well-documented difficulties in receiving evidence-based care. On average, only 23.2% of individuals with a diagnosable ED seek treatment [51]. Treatment seeking appears to be a pronounced issue among younger individuals [51,52]. Rates of treatment seeking amongst adolescent females with symptoms of an ED have been reported to be as low as 2.6% [53]. There is also evidence to suggest a mean delay of 5.28 years between the onset of ED symptoms and the first instance of help-seeking behaviours, with this figure being significantly longer for individuals with BN (8.40 years) as compared to other EDs (e.g., AN, 2.20 years) [54]. This is of particular concern given the well-evidenced importance of early, targeted treatment in offsetting the risk of increasingly complex and chronic ED presentations [15]. Additionally, for those individuals who do reach out to access services, most do not seem reinforced for their efforts. A report by National Eating Disorders Collaboration (NEDC) [55] found that up to 85% of the Australian population seeking ED treatment experience difficulty in accessing an appropriate treatment option. A major obstacle in the enhancement of mental healthcare is the tendency for the updating of clinical practices to lag behind research-driven developments, otherwise known as the “*research-practice gap*” [56,57].

### 2.1. Dominant Model of Treatment Delivery

A major contributor to the research–practice gap is the widespread promotion of individual, therapist-led psychological treatments as the primary model of treatment delivery [58] (Figure 1). Three main features of this dominant model of psychotherapy limit its scalability: (1) treatment sessions delivered in a one-on-one format, (2) highly trained health professionals required for dissemination, and (3) sessions delivered in-person at a clinic or healthcare facility [59]. With such stringent requirements, the availability of skilled clinicians is narrowed considerably, rendering the healthcare system incapable of managing existing service demand [60,61]. This model is considered inflexible not only for clinicians but also for patients as it requires a substantial time commitment on behalf of the patient to both travel to and attend regular face-to-face therapy sessions. This challenge is only made more difficult for individuals who are engaged in full-time employment or study, for whom attending appointments within business hours is nearly impossible [62].

### 2.2. Workforce Limitations

The NEDC (2014) report found that demand for specialist ED outpatient services in Australia is high, meaning that waitlists are common and therapist time is a precious resource [55]. Under-resourced and under-staffed clinics are limited in their capacity to deliver the evidence-based 20-week cognitive behaviour therapy (CBT) treatment program, requiring a minimum of 20 h of therapist contact. Additionally, clinicians in non-ED specialist settings are often unable to deliver the full continuum of care to ED patients as very few staff members are trained in ED treatment, resulting in patients receiving unspecific care rather than evidence-based treatments for their illness [56,63].

Concerns regarding workforce limitations across the Australian health system have become more pronounced in response to the coronavirus disease 2019 (COVID-19) pandemic. As part of a survey conducted from January to February 2022, the Australian Psychological Society found that approximately one third of psychologists are not taking on new clients due to overwhelming demand [64]. This is an increase from only 1% of psychologists who were unable to see new clients prior to the pandemic. The Royal Australian and New Zealand College of Psychiatrists has described the workforce shortages as “extreme” and “dire” as lengthy wait times, poor distribution, and a lack of services plague the field [65].

### 2.3. Geographical Barriers

Workforce limitations are substantially worsened in regional and rural areas in which scarcity of empirically supported psychotherapy is high [66]. Mental health professionals tend to be concentrated in highly populated urban cities, rendering residents of smaller, non-metropolitan towns deprived of specialist psychiatric care and even less likely to access clinicians trained in ED treatment [56]. There is evidence to suggest that very few to no specialist ED services operate within regional areas of Australia, necessitating the far majority to travel to metropolitan areas to access specialist care [44]. Inequality exists not only in service provision, but also in mental health status with evidence to suggest that individuals living in rural communities experience poorer mental health than residents of metropolitan areas [67].

### 2.4. Health System Barriers

Uptake of ED treatment also varies across populations and countries, with some barriers systematically embedded within the broader healthcare system in place. In Australia, and most other Western countries, a significant challenge is the provision of treatment services across the full continuum of symptom severity of BN [66]. According to the Productivity Commission’s Inquiry into Mental Health (2020), this is of particular concern for individuals with moderate to severe mental health difficulties who appear to be slipping through the gaps of care [68]. Such individuals are referred to as the ‘missing middle’, as they are not considered to be severely unwell enough to access state-funded acute and continuing care (e.g., inpatient units, community mental health), however they are also deemed too complex to be treated by primary care services (e.g., GP or counsellors). As a result, such individuals are often forced to rely upon the private health sector, which for the far majority, is a significant expense.

### 2.5. Cost of Treatment

The cost associated with private health treatment can be exponential and, depending on the funding initiatives available within the given country, patients are only likely to receive financial support for a portion of the expense encountered. The average treatment cost is AUD 6000 for each individual case of BN in Western countries [69]. For Australian residents, a government-funded initiative introduced in November 2019 enables individuals with an ED to access up to 60 subsidised sessions with health professionals (40 psychological, 20 dietetic) [70,71]. Despite the financial support offered by this initiative, ED-specific psychotherapy is a specialised treatment such that trained clinicians are inclined to charge a premium fee, resulting in a price gap between the rebate and fee which remains unaffordable for many. There is evidence to suggest that approximately 60% of patients experience out-of-pocket fees for Medicare-funded psychological treatments [72]. The cost of treatment represents a significant barrier to accessing care, especially for young individuals, with the Australian Bureau of Statistics reporting that 20% of young Australians in need of mental healthcare do not access support from psychologists or psychiatrists due to the cost associated with these services [72,73,74].

### 2.6. Fidelity of Treatment

In the event that geographical and financial barriers are overcome, the fidelity of CBT is often compromised by ‘therapist drift’ which has been identified as a prominent issue across several evidence-based treatment programs [75]. Therapist drift occurs when a clinician deviates from evidence-based treatment protocols. Therapist drift is thought to contribute to the discrepancy between the outcomes achieved in clinical research trials and routine clinical practice [76]. The quality of CBT being delivered by clinicians in the community is at best questionable, with studies reporting that as few as 6% of clinicians adhere to evidence-based CBT manuals [77]. This calls into question whether ED patients are receiving the manualised care proven to be effective in clinical trials.

### 2.7. Stigma and Shame

In addition to service-side barriers, there are patient-associated factors which hinder help-seeking. One of the most prominent barriers reported by patients is the shame and stigma associated with accessing mental healthcare [63]. Fear of stigmatisation appears to be compounded for individuals with BN given the tendency for EDs to be perceived as a “lifestyle choice” rather than an illness, in addition to the shame and embarrassment associated with binge-eating behaviours. Additional barriers associated with the social and emotional psychopathology of EDs include low motivation to change, denial, or failure to perceive the severity of the illness [15,63,78,79].

## 3. Treatment of Bulimia Nervosa

There is a discrepancy between what is collectively *known* about effective treatments for BN and what is *accessible* to individuals in need of care. It is well established that the receipt of care is compromised for individuals with BN; however, this stands in contrast to the range of psychological therapies that have been evaluated for the treatment of BN. CBT is recognised as the first-line treatment for adults with BN [80] (see Section 3.1). In addition to CBT, several other therapeutic approaches have received attention from researchers as potential treatment options [81,82]. This section will first provide an overview of the available evidence regarding the use of interpersonal psychotherapy (IPT), dialectical behaviour therapy (DBT), psychodynamic psychotherapy, and pharmacotherapy in the treatment of BN, followed by a detailed review of the evidence base supporting CBT.Interpersonal Psychotherapy

IPT attempts to address an individual’s interpersonal skills, including dispute resolution, problematic interactive styles, and communication training, all of which are thought to be involved in the maintenance of maladaptive coping styles characteristic of BN [83]. IPT is considered to be a strong, evidence-based alternative to CBT, despite holding a lesser degree of empirical support than CBT [80,84]. Research trials comparing CBT to IPT suggest that CBT results in greater abstinence rates post-treatment as compared to IPT; however, significant differences in outcome were no longer apparent at one-year follow-up [85,86]. The studies emphasise that although CBT and IPT produce similar long-term improvements, temporal differences in treatment response are present such that IPT has a protracted, slower course and requires a longer period of time to achieve similar effects to CBT. However, some recent research by the team behind CBT-E suggests that the ‘enhanced’ form of CBT (CBT-e; see Section 3.1.3) maintains superior treatment outcomes over IPT even at follow-up [87].Dialectical Behaviour Therapy

DBT is a type of cognitive behaviour therapy originally developed for the treatment of BPD that has been adapted to address behaviours of binge eating and compensation in BN [88,89]. According to a DBT framework, emotion dysregulation is considered to be the core problem in BN, such that disordered eating behaviours are conceptualised as dysfunctional attempts to regulate painful emotional states [90]. DBT aims to equip individuals with a range of skills to help modulate the expression and experience of an emotion and therefore reduce reliance upon dysfunctional behaviours. Strategies taught to patients as part of a DBT intervention include emotional regulation skills, distress tolerance, mindfulness, and self-soothe activities [90]. Current empirical evidence indicates that DBT treatments produce significant reductions in disordered eating behaviours at post-treatment in comparison to waitlist control (WLC). However, the evidence base is relatively weak, mainly consisting of small sample sizes and very few studies assessing the maintenance of improvements at follow-up [82,91].Psychoanalytic Therapy

Much less frequently evaluated treatments include psychodynamic and psychoanalytic approaches [80]. Psychoanalytic psychotherapy is typically delivered via weekly sessions across two to three years and therefore spans a substantially longer treatment period in comparison to other briefer psychotherapies such as CBT [92,93]. The therapeutic targets of psychoanalytic psychotherapy are markedly different to other more behaviourally focused therapies in that the therapy adopts a nondirective approach to exploring the relationships between the unconscious mechanisms thought to trigger the bulimic symptoms [94]. A randomised controlled trial (RCT), comparing psychoanalytic psychotherapy to CBT-e in the treatment of BN, demonstrated that CBT-e was significantly more effective than psychoanalytic psychotherapy in alleviating binging and purging symptomology, and was able to do so within a much shorter time period [94].Pharmacological Treatments

Pharmacological treatments, in particular antidepressant medications, have been shown to be effective in reducing binge eating frequency by 50–70% for individuals with BN [80,95]. Given the high comorbidity between BN and mood disorders, it is hypothesised that the effectiveness of antidepressant medications in this clinical population is indicative of an association between depressive symptoms and control of one’s eating behaviours [96,97]. The stimulant medication lisdexamfetamine dimesylate (LDX), otherwise known as Vyvanse, has also received growing attention in the treatment of BN. LDX has been proven effective in adults with moderate to severe binge eating disorder (BED). Empirical support stemming from four RCTs has demonstrated superior treatment outcomes in terms of binge episode frequency for BED patients after receiving LDX as compared to a placebo [98]. To date, there are no published RCTs examining the use of stimulant medication in BN; however, preliminary data from an open-label two-month feasibility trial of LDX demonstrated a significant reduction in objective binge episodes and compensatory behaviours, with large effect sizes reported by authors [99]. Overall, clinical guidelines recommend against the use of pharmacotherapy as an independent treatment approach for individuals with BN, with empirical evidence suggesting that drug-alone trials are characterised by higher attrition rates and lower binge-eating abstinence rates as compared to medication treatments combined with CBT [95].

### 3.1. Cognitive Behavioural Therapy

Despite the existence of several therapeutic approaches for BN, CBT upholds the most compelling evidence base. A recent meta-analysis of RCTs of CBT treatments for EDs found therapist-led CBT for BN to be more efficacious than active comparator treatments in improving behavioural and cognitive symptoms for participants, reporting small to large effect sizes [84]. The National Institute for Clinical Excellence (NICE) recommends BN-focused CBT as the leading treatment for BN, with a strong evidence base to support its efficacy in reducing disordered eating behaviours and related cognitions both in the short term and long term [80,81,84,100,101,102,103].

#### 3.1.1. Cognitive Behavioural Model of Bulimia Nervosa

The cognitive behavioural model of BN was developed by Fairburn, Cooper, and Cooper (1986) as an extension of existing models of AN and in an attempt to better understand the origin and maintenance of the illness. The model suggests that over-evaluation of weight and shape sits at the core of bulimic psychopathology and is the driving mechanism behind extreme caloric restriction and strict adherence to dietary rules [104]. Binge-eating episodes are thought to be prompted by chronic hunger caused by dietary restriction which leads the individual to overeat in compensation or in response to a minor deviation from one’s self-imposed rules. This is interpreted by the individual as a personal failure or sign of ‘lack of control’, causing them to relinquish control and engage in binge eating. The binge episode is then followed by repeated use of extreme weight-control behaviours, driven by a prevailing fear of weight gain. The model suggests that it is the recurrent binge–compensation cycle which serves to strengthen the individual’s obsessive body-related concerns and repeated attempts to maintain self-control [104]. The model also suggests that negative events or adverse moods make an individual more vulnerable to binge episodes, with low-mood states likely to create greater difficulty in adhering to stringent dietary restriction [105]. Further, Fairburn et al., (2003) proposed that binge eating may function as a form of mood regulation, serving to modulate negative emotional states [105].

#### 3.1.2. Cognitive Behavioural Treatment for Bulimia Nervosa

Shortly following the addition of BN to the DSM-3 [7], an illness-specific, therapist-led version of CBT was developed for BN by Fairburn and colleagues (CBT-BN) [106]. CBT-BN is a manualised outpatient treatment, consisting of 20 individual sessions delivered over 5 months [107]. As part of CBT-BN, individuals are guided by a mental health clinician through a series of behavioural and cognitive techniques intended to address their dysfunctional self-evaluation mechanism. Three overlapping phases form part of the treatment: (1) psychoeducation about BN and its maintenance, initiation of self-monitoring of food intake, and introduction of regular eating; (2) skills to address maladaptive cognitions and behavioural experiments aimed at reducing avoidance behaviours; and (3) strategies to maintain improvements (Table 2).

CBT-BN has been found to result in complete cessation of binge eating and purging behaviours in approximately 40 to 50% of treated patients CBT-BN [85,108,109]. CBT-BN produces superior treatment outcomes in comparison to inactive controls, other short-term psychological treatments (including IPT, exposure, response-prevention treatment, behaviour therapy, and short-term focal psychotherapy), and antidepressant medications [104,110,111,112]. The therapeutic effects of CBT-BN also appear to extend to general psychopathology, including improvements in depressive symptoms, self-esteem, and social functioning [113]. In terms of predictors of treatment outcome, lower impulsivity and shape concern have been associated with better treatment outcomes; however, inconsistent findings across studies limit the degree to which reliable predictors and moderators can be identified [114,115].

A noteworthy study conducted by Fairburn and colleagues (1993) compared CBT-BN to behaviour therapy and IPT in a sample of 99 patients with BN. Overall, the three treatments produced substantial and sustained improvements in ED psychopathology, as measured using the global Eating Disorder Examination (EDE) score; however, treatment outcomes for participants who received CBT showed lower overall symptoms at six-year follow-up compared to behaviour therapy or IPT [86]. Similar findings were replicated in an RCT conducted by Wilson et al. (2002) which compared CBT-BN and IPT using a sample of 220 participants who met the DSM-3 criteria for BN. Following 19 individual sessions of treatment across a 20-week period, CBT significantly outperformed IPT in the reduction in both binge-episode frequency (80% for CBT, 52% for IPT) and purging frequency (80% for CBT, 44% for IPT). The authors also found CBT-BN to be associated with a more rapid treatment effect than IPT, with 62% of post-treatment improvement apparent by the sixth week for patients who received CBT-BN [109]. 

#### 3.1.3. Enhanced Cognitive Behavioural Therapy

More recently, Fairburn reformulated the CBT-BN treatment model to account for a greater breadth of ED psychopathology. Rather than being a specific treatment for BN, the novel treatment program is guided by the transdiagnostic model of EDs, which accounts for the spectrum of ED diagnoses [116]. Enhanced CBT or CBT-e aims to address the core features of ED psychopathology, namely overvaluation of weight and shape and dietary restraint, in a structured, therapist-led four-stage program, which takes place across a 20-week period [87,117]. It also includes modules to address processes “external” to the ED that are thought to interact with the core ED psychopathology, such as clinical perfectionism, core low self-esteem, interpersonal difficulties, and mood intolerance. Several systematic reviews examining CBT-e have concluded that it is an efficacious treatment for patients with BN [84,118,119,120]. A recent meta-analysis of CBT-e by Atwood and Friedman (2020) reported that abstinence rates from binge eating and purging in the past 28 days ranged from 22.5% to 44% at post-treatment, with these rates being well maintained up to 19-month follow-up [121].

For patients with BN, CBT-e outperforms delayed treatment control groups and active comparison treatments, including IPT and psychoanalytic psychotherapy [122]. However, there is no evidence to suggest that CBT-e is more effective than CBT-BN for individuals with BN. No research studies have directly compared CBT-e and CBT-BN and, of the available evidence examining CBT-e, remission rates are comparable to those produced by CBT-BN both at post-treatment and follow-up [121]. Additionally, Linardon et al.’s (2017b) meta-analytic review of CBT treatments for EDs found no differences in effect sizes between studies using CBT-BN and CBT-e protocols [84].

#### 3.1.4. Clinician-Supported and Pure Self-Help Cognitive Behavioural Therapy

While there is strong empirical support for the use of therapist-led CBT-BN in the treatment of BN, there is also good evidence to suggest that similar outcomes can be achieved using less-intensive CBT-based interventions, involving much briefer interactions with clinical support [84,123]. According to this approach, individuals have traditionally been guided through a standardised written program or manual containing educational material and skills related to the core effective components of CBT [123,124,125]. Such interventions, broadly characterised as self-help treatments or manuals, are designed in such a way that they can either be delivered in a purely independent, self-directed manner (pure self-help) or in conjunction with support from a mental health professional (guided or supported self-help). Fairburn’s *Overcoming Binge Eating* is based upon the core principles of CBT-BN and is one of the most widely researched self-help manuals for BN [126]. Other evidenced-based self-help manuals for bulimic and binge-eating phenotypes include *Bulimia Nervosa and Binge Eating: A Guide to Recovery* [127] and *Getting Better Bite by Bite: A Survival Kit for Sufferers of Bulimia Nervosa and Binge Eating Disorders* [128]. Clinician-supported self-help is distinct from traditional psychological therapy in terms of both the content and structure of the clinical encounter. The primary aim of clinician-supported self-help is to facilitate the patient’s engagement with the therapeutic strategies being delivered within the self-help manual [129]. It involves much less direct clinical contact than traditional clinician-led treatments, with most programs consisting of a maximum of 12 sessions and an average length of 20 min each [130].

Although pure self-help CBT treatments have been demonstrated to be effective in reducing binge and purge frequency both at post-treatment and at follow-up assessment [123,131,132,133], most studies show that significantly improved treatment outcomes are achieved through the addition of regular clinical support [81,124,134]. In a trial examining Fairburn’s (1995) *Overcoming Binge Eating* self-help manual, symptom reduction increased from 25%, for participants engaging in the treatment independently, to 36% for those receiving 30 min monthly clinician support sessions via telephone, to 50% for those who received support in a face-to-face format [135]. There is also research demonstrating that abstinence from bulimic behaviours following a supported self-help program can be maintained for up to 12-months post-treatment [136].

CBT delivered in a guided self-help format has been shown to outperform both WLC and active psychological treatments in the reduction in the core behavioural and cognitive features of BN [123,137,138]. Studies directly comparing clinician-supported self-help CBT to another psychological intervention report that more rapid symptom reduction and significantly higher remission rates are achieved in response to guided self-help interventions (e.g., 74% for clinician-supported self-help vs. 44% for in-person group CBT; Bailer et al., 2004). There is also evidence to suggest that clinician-supported self-help treatments can perform just as well as clinician-led CBT. Mitchell and colleagues (2011) conducted one of the largest controlled studies to date examining a stepped-care model approach for the treatment of BN, comparing standard face-to-face, clinician-led CBT-BN delivery and clinician-supported self-help [139]. The trial set out to compare a full course of individual, therapist-led CBT (20 sessions, 50 min length) with clinician-supported use of Fairburn’s (1995) *Overcoming Binge Eating* manual (8 sessions, 20 min length), across an 18-week treatment period. Post-treatment measures indicated no significant difference between remission rates for participants who received traditional CBT (57%) and clinician-supported self-help CBT (52%). In a similar study comparing clinician-supported self-help and therapist-led CBT in a sample of 62 patients with BN, Thiels et al. (2003) found almost equivalent abstinence rates between the two groups in terms of binge eating, purging, and laxative use at four-year follow-up (66.7% for clinician-supported self-help, 61.5% for therapist-led CBT). Taken together, these findings have led researchers to conclude that clinician-supported self-help programs represent an effective first-line treatment for BN [139]. Empirical support for the efficacy of CBT-guided self-help in this illness group has translated into recommendations by four of the seven international guidelines for ED treatment to use CBT-based guided self-help interventions in the treatment of BN [61].

In addition to being an efficacious treatment option, self-help interventions accommodate for delivery of the effective components of treatment in a manner that places lower demands on resources. Clinician-supported self-help treatment requires one fifth of the therapist contact hours needed for a complete CBT course, freeing up a greater amount of time for specialist therapists to devote to more chronic, complex, or treatment-resistant patients [85,133,140]. Additionally, supported self-help can be guided by a practice nurse, primary care mental health worker, or intern, as opposed to the higher level of training from a clinical psychologist or psychiatrist required for individual therapy [141].

Due to the scalable and cost-effective nature of self-help interventions (supported and pure self-help), researchers and policy makers have endorsed these treatments as being well suited to a stepped-care model of healthcare delivery [142]. Stepped-care models attempt to maximise the key limiting factor in most publicly funded healthcare systems: specialist clinician time [143,144,145]. As such, briefer, low-intensity treatment options are offered to individuals with less-severe clinical presentations in order to reserve specialist therapist time for those with more complex symptomology who may not respond to simpler first-line treatments. As part of the model, systematic monitoring of patients’ treatment progress is required so that, if clinically indicated, a “step up” of treatment intensity can occur in order to ensure that the low-intensity options are not counterproductive and leading to a worsening of symptoms [143,146]. The NICE 2017 guidelines recommend that guided, CBT-based self-help programmes be adopted as a first-line treatment for BN with instruction to assess progress at week four of treatment and ‘step up’ to clinician-led CBT if self-help is considered ineffective [102]. In addition to being a time-effective approach, stepped-care models have also been shown to produce a more cost-effective system of healthcare delivery. Crow et al. (2012) estimated the total cost of treatment for BN within a stepped-care framework to be USD 12,146 per patient, a significantly reduced cost as compared to routine, individual CBT estimated to cost USD 20,317 [147].

## 4. Digital Mental Health Interventions

As reviewed in the earlier discussion of barriers to treatment, notable challenges exist in the dissemination of evidence-based care to those most in need. Digital platforms have emerged as a powerful modality to overcome several of the existing health system barriers, especially those associated with traditional face-to-face treatment [148]. Digital mental health interventions broadly refer to the delivery of evidence-based psychological strategies via information and communication technologies, such as a computer, mobile phone (smartphone), or tablet [149]. There is an important distinction to be made between the delivery of clinician-led treatments via digital means (e.g., client–therapist session via videoconference) and asynchronous, Internet-based self-help programs. This review will focus on the latter.

Much like with written self-help manuals, digital self-help programs or eTherapies consist of a structured format, typically 30 to 45 min modules per week for six or more weeks, and are often adapted from evidence-based models, most commonly CBT manuals. The delivery of therapeutic content is via digital multimedia channels (e.g., text, video, audio), organised into interactive online modules to be completed in a self-directed manner from week to week [150]. Research suggests that engagement with self-help treatments is closely related to the way in which treatment is accessed (i.e., via computer, book, or the Internet). Specifically, dropout rates for self-help interventions delivered via bibliotherapy are reportedly twice as high as Internet-based interventions [151,152]. Accordingly, the field of digital interventions has attracted researchers in the pursuit of interactive and engaging delivery channels for self-help treatments in favour of book-based interventions.

A key advantage of digital interventions is their reach and ease of access. Between 2017–2018, 86% of Australian households reported having access to the Internet and 91% of these households endorsed using computers or smartphones to access the Internet [153]. The widespread use of technological devices facilitates access to evidence-based treatments in a manner that is more flexible and cost-effective for the patient [154]. Digital platforms provide scope for mental healthcare to be uniformly delivered to numerous users simultaneously, offsetting the delay associated with long waitlists for face-to-face psychotherapy [150]. The unrestricted geographical reach of digital interventions allows for national healthcare service delivery to be extended into rural and regional areas. Digital treatments are also a highly cost-effective method of treatment delivery and can ensure that users receive access to a standardised, evidence-based dose of treatment, with a potential to eliminate issues such as therapist drift [58].

The technological capabilities of digital interventions can be harnessed to create tailored, personalised interventions. Sophisticated features such as real-time symptom monitoring, collection of electronic trace data (e.g., log-in rates, time spent using intervention), GPS tracking, and machine-learning algorithms can be used to assess and track both illness severity of the user and their degree of treatment engagement in order to tailor the type or intensity of the intervention [155,156]. Approximately 91% of smartphone owners report not leaving their homes without their phones and 46% of owners state that they could not live without their phone [157,158]. Such statistics reflect the powerful capacity for digital devices to create and maintain strong habits and, more importantly, demonstrate the potential for such devices to be applied as part of behaviour-change interventions [159].

From a patient perspective, the largely anonymous nature of technology-based interventions may appeal to the high levels of shame and pervasive secrecy experienced by this clinical population regarding their bulimic behaviours [160,161,162]. Additionally, the flexibility offered by digital interventions means that users are able to engage in therapy at any time and location that best suits them, helping patients to overcome logistical barriers of attending face-to-face appointments [56,58].

### 4.1. Digital Interventions for Eating Disorders

A growing evidence base for digital mental health interventions is apparent in ED research. The field has experienced a rapid increase in the number of trials evaluating the use of technologies in ED treatment from basic tools such as CD-ROMs through to sophisticated eTherapy technologies, including virtual reality and digital self-monitoring tools [163]. The target audience for digital interventions is widespread and spans the illness trajectory, including prevention and early intervention programs, guided self-help treatments, relapse prevention, and maintenance of treatment gains, as well as carer support programs [58]. It appears that the most extensive research has been conducted on prevention-focused interventions, with 28 of 36 RCTs included in a recent systematic review categorised as prevention-focused and only 8 categorised as treatment-focused [164].

#### 4.1.1. eTherapy Programs for Eating Disorders

There is empirical evidence to support the efficacy of eTherapy treatment packages for EDs [164]. The majority of eTherapies for EDs are based upon CBT, 8 to 12 weeks in length, delivered in a series of treatment modules with a pre-defined duration for completion (most commonly seven days) [149,165]. Several meta-analytic reviews have been conducted in recent years in an attempt to synthesise findings from the growing number of research trials in the field [95,148,149,165,166,167,168,169]. Most reviews conclude that treatment-focused trials of eTherapy programs produce a significant reduction in behavioural and attitudinal symptoms of EDs in comparison to control conditions. Moderate to large effect sizes have been reported for changes pre- to post-treatment and pre-treatment to follow-up for outcome measures including binge/purge behaviour, ED psychopathology, shape/weight concerns, and dietary restraint (Hedges’ *g* range from 0.51 to 1.06) [149,164,165]. Improvements in comorbid mental health symptoms, including depression, anxiety, and quality of life, have also been reported across several studies for participants receiving an eTherapy intervention as compared to control groups [165]. Despite such promising findings, several reviews have expressed caution regarding the interpretation of these results. They warn that only a small number of RCTs (*n* = 6–8) with sound methodological quality have contributed to these effects and that the use of mixed clinical populations (i.e., multiple ED diagnoses) across studies potentially limits the reliability of the findings, as well as the ability to generalise results across ED diagnoses [95,164,166].

Unlike the broader literature regarding written self-help interventions, it is unclear whether the addition of clinician support alongside online ED interventions produces larger treatment effects as compared to pure self-help treatments [167]. Further research is required to determine the exact strength and nature of the association between clinician support and outcome, with reviews noting that little variability in the design of trials within the field has resulted in limited knowledge about the impact of factors such as the frequency and modality of contact (e.g., email, phone call, videoconferencing) on treatment outcomes [165,166]. Aardoom and colleagues (2016) conducted one of the only studies in the ED literature which has attempted to gain further clarity on the role of clinician support in ED online treatments. As part of an RCT, they delivered a weekly, eight-session online program (*Featback*) to a sample of 354 patients diagnosed with self-reported ED symptoms [170]. In addition to the online program, participants were randomised to treatment conditions with varying levels of therapist support delivered via digital means (i.e., email, online chat, and/or videoconference). The levels included no clinician support, contact once per week, and contact three times per week. A significant reduction in bulimic psychopathology, as well as symptoms of depression and anxiety, were reported across all three arms, with improvements shown to be superior to WLC [170]. No additional benefits of therapist support were observed in terms of symptomatic improvement; however, it did lead to greater participant-reported satisfaction with the intervention as compared to those who received no therapist support [170].

In terms of attitudes towards digital interventions by their intended users, the data are mixed. McClay and colleagues (2016) assessed attitudes towards online treatments using a sample of individuals recruited to participate in a trial of a digital ED intervention [171]. It was found that an overwhelming majority of participants (98%) held positive views towards eTherapies, with 78% reporting a preference for clinician support alongside the intervention. Participants were also asked to express their concerns regarding digital interventions; confidentiality (24%) and privacy (17%) were the only reported concerns. Linardon et al. (2020) similarly assessed preferences towards digital interventions in a community sample of participants with varied ED symptomology. The majority of participants reported a preference for face-to-face therapy with only 27.5% of participants preferring eTherapies (of these, 19.2% favoured clinician-supported and 8.3% favoured pure self-help). Despite this, half of the participants reported an intention to use eTherapy programs for current or future disordered eating behaviours. The type and severity of ED symptomology were not associated with attitudes or preferences towards digital interventions, a finding that is inconsistent with the hypothesis that lower-intensity treatment options may be viewed as insufficient by individuals with severe symptoms [172]. Overall, it appears that even with preferences relating to face-to-face delivery withstanding, potential users of digital interventions are likely to respond favourably to the availability of low-intensity treatment options.

#### 4.1.2. eTherapy Programs for Bulimia Nervosa

Research on digital interventions for EDs has primarily focused on the treatment of binge-eating and bulimic phenotypes (i.e., BN, BED, or OSFED with binging or bulimic symptoms) with very few studies evaluating the use of digital interventions for AN. It is thought that attempts to incorporate digital modalities into the treatment of AN have been limited by the severity of condition, prioritising caution and accessibility compared to intensive face-to-face options [164]. The evidence supporting the use of online interventions is strongest for BN [164]. Moderate-pooled effect sizes (Hedges’ *g* range from 0.63 to 0.73) have been found across trials comparing CBT-based eTherapy for BN to WLC for improvements to binge eating, ED psychopathology, and dietary restraint from pre- to post-treatment. Results pertaining to changes in purging frequency are less consistent across studies [149,164,173].

Sánchez-Ortiz et al. (2011) reported on the effectiveness of an eight-session online CBT-based program in an RCT of 75 full-threshold and subclinical BN patients [173]. They found that 25.8% of patients who engaged in the eTherapy program displayed abstinence from bulimic behaviours at post-treatment measures, which is comparable to the 20% and 30% abstinence rates reported in other trials of manual-based CBT self-help and face-to-face CBT, respectively [133,174]. Moreover, these improvements were maintained at six-month follow-up and even improved further for some individuals, with 52.2% of participants who completed the online CBT intervention no longer meeting criteria for an ED at follow-up. The effectiveness of online CBT for BN may also extend to the treatment of adolescents. A non-randomised study by Pretorius et al. (2009) reported significant improvements in objective binge episodes, self-induced vomit episodes, and global EDE score at post-treatment and three-month follow-up in a sample of 101 participants aged 13 to 20 years with BN [175].

Two additional RCTs have compared the effectiveness of online CBT to another active intervention, namely a conventional bibliotherapy intervention, in the treatment of BN [176,177]. Ruwaard et al. (2013) recruited 105 participants with varied levels of bulimic symptoms who were then randomised to receive either a ten-session CBT-based online program or a hard copy of a self-help book for BN to complete across a 20-week period [176]. Both intervention groups reported a significant reduction in Eating Disorder Examination–Questionnaire (EDE-Q) global score, binge-eating frequency, purging frequency, and the Body Attitude Test (BAT) total score. Online CBT outperformed bibliotherapy in terms of pre- to post-treatment reductions in ED symptoms; however, the differences between the two interventions were no longer significant at one-year follow-up as a result of further symptom improvements in the bibliotherapy group. Wagner et al. (2013) reported similar findings in a large RCT (*n* = 155) comparing online CBT to bibliotherapy *Getting Better Bit(e) by Bit(e),* [128]. The interventions were available for participants to use across a four to seven month period and were delivered alongside support from an ED specialist clinician in the form of a weekly email to the participant. Both intervention groups reported a significant decrease in the frequency of objective binge episodes, self-induced vomiting, laxative misuse, excessive exercise, and fasting from pre- to post-treatment, with the greatest reduction within four months, followed by further improvement at seven months, and maintained improvement at 18-month follow-up. However, contrary to the hypothesised superiority of online CBT, no differences in outcome were found between the two treatments.

Again, the number of studies contributing to the effect sizes reported in reviews to date are low, with only a handful of controlled trials having examined the clinical effectiveness of online self-help treatments in BN [173,175,176,177]. While some reviews claim that eTherapies may be a good alternative to face-to-face interventions [149,165], other express concern regarding a high degree of inconsistency and methodological weakness in existing studies (e.g., varied outcome measures, small sample sizes, high risk of bias, and heterogeneity). Accordingly, researchers have been called upon to produce high-quality studies in this area in an attempt to strengthen the evidence base [95,164,166].

#### 4.1.3. Smartphone Applications

Digital self-help programs have been typically designed for delivery via a computer; however, there has been growing interest in the use of smartphone applications (“apps”) as a more accessible and engaging mode of delivery [156]. Adherence is a well-documented drawback of digital self-help programs. App-based interventions offer the opportunity to enhance real-world engagement with data, suggesting increasing rates of smartphone usage since their release [178]. Smartphone technology has become profuse in daily life, such that users have been found to use their phones an average of 84 times per day, totalling approximately five hours daily [178]. App-based digital mental health interventions have been shown to be efficacious for a range of psychological disorders, including mood disorders, anxiety disorders, and psychosis prevention [179,180,181,182]. To our knowledge, none of the empirically supported eTherapies for EDs discussed in Section 4.1.1. are publicly available on the commercial marketplace or exist as a smartphone app. Despite this, a plethora of ED apps are available to the public via app stores (e.g., iOS and Google Play stores). A recent review identified 65 apps which are publicly available to support the treatment of EDs and found that less than 10% of these have any research behind them [183]. This raises concern that the access point to digital mental health interventions for individuals seeking care is dominated by a series of unregulated ED app developers with no association with ED treatment centres, hospitals, or academic universities. Although some reviews have reported that most apps include components of evidence-based treatments, there is a high risk that as a package they may be therapeutically ineffective or, at worst, deliver inaccurate or harmful information and interventions [183,184,185].

Among the published research studies on ED-focused apps, most focus on a narrow range of apps. Two prominent smartphone apps have been made available for the self-monitoring of ED behaviours, namely *Recovery Record* and *Noom Monitor*. Self-monitoring is considered to be one of the most powerful therapeutic components of CBT for BN, with studies reporting on its unique effectiveness in reducing binge-eating behaviours in CBT treatments [186,187,188]. Both *Recovery Record* and *Noom Monitor* have adopted a digitalised version of original pen-and-paper CBT-based self-monitoring records, including fields for users to record their clinically relevant behaviours, such as food and drink intake, binge-eating episodes, and compensatory behaviours, and associated features such as thoughts and mood [189,190]. In addition to the patient self-monitoring app, *Recovery Record* and *Noom Monitor* also have an adjunct clinician app to allow for joint monitoring of the patient’s progress if used in a clinician-supported format. Technological devices uphold several features with the potential to improve the convenience of self-monitoring, including inbuilt reminders to prompt higher engagement and immediate personalised feedback based upon users’ entries [191]. Additionally, digital self-monitoring platforms via smartphone app may improve the accuracy of real-time monitoring, posing a more convenient and anonymous alternative to pen-and-paper monitoring.

A 2014 case report carried out by Tregarthen and colleagues suggests that clinicians and patients have readily embraced the *Recovery Record* app, with data noting 108,000 downloads of the app in the first two years of its launch on the commercial marketplace [190]. A follow-up RCT was recently conducted comparing two versions of Recovery Record, including a standard version (i.e., self-monitoring only) and a newly developed tailored version (i.e., self-monitoring plus personalised CBT content). Participants (*n* = 956) were randomised to either study group to engage with the app for eight weeks. Both the standard and tailored group displayed clinically meaningful change on the EDE-Q (61.6% and 55.4%, respectively); however, no significant difference was observed between randomised groups [192].

Similarly, *Noom Monitor* has been evaluated in two RCTs as an adjunct tool to a clinician-supported self-help treatment [189,193]. As part of Hildebrandt and colleagues’ most recent study, participants with BN or BED were randomised to receive standard care (i.e., non-ED specific, generalised psychiatric service) or a 12-week intervention consisting of the *Noom Monitor* app in addition to a CBT self-help manual and eight telemedicine clinician support sessions [189]. Participants who engaged in the *Noom Monitor* treatment group reported significant reductions in objective binge days and achieved higher rates of remission from binge eating at 52-week measures when compared to standard care (56.7% and 30%, respectively).

A recently developed digital ED intervention, known as the *Break Binge Eating* program, also includes a smartphone self-monitoring app [194]. The *Break Binge Eating* program consists of a blended Internet and app-based intervention, which together solely target dietary restraint as the mechanism to reduce binge episodes. The psychoeducation component of the intervention is delivered via the Internet and the app allows users to monitor their behaviours using the digital diary as well as complete between-session exercises. The program was examined in a recent RCT using a sample of 403 participants who registered themselves for the trial [195]. There were no eligibility criteria related to ED symptom level; rather, the researchers adopted the assumption that most individuals expressing interest would have some degree of elevated dietary restraint or binge eating. At baseline, 90% of participants reported having experienced at least one binge episode per week in the past month. Participants were randomised to either receive the *Break Binge Eating* intervention or an informational control group. Linardon et al. (2021) found that participants in the intervention group displayed significantly greater reductions pre- to post-treatment in the frequency of objective binge episodes as compared to the control group (small effect size, Risk Ratio = 0.60) [195]. Significant reductions were also observed in secondary outcomes including the EDE-Q subscales of shape concern, weight concern, and eating concern as compared to the WLC group. Additionally, such improvements were maintained at an eight-week follow-up assessment.

## 5. Engagement and Adherence

A well-documented issue in trials examining online treatments is the high rates of participant dropout [196,197,198]. A 2018 meta-analytic review of CBT programs for Eds (*n* = 99) reported that the highest dropout rates of all modalities were observed in studies that delivered Internet-based CBT (range: 20% to 36%). Other reviews of RCTs of digital ED interventions have reported average dropout rates of 25.3% up to 47.2% [164,165]. This challenge is not unique to digital interventions for EDs, with trials of other psychiatric conditions reporting similar difficulties with maintaining user engagement. A review of 19 studies assessing Internet-based treatments for psychological disorders reported a weighted average dropout of 31% (range: 2–83%) across studies, raising uncertainty regarding the dose of treatment that participants are in fact receiving [199]. An updated review of digital self-help interventions for depression or anxiety (*n* = 11) by Fleming et al. (2018) found that the number of users engaged with the intervention decreased substantially after the midpoint of treatment. Specifically, between only 7 and 42% of users were considered to have engaged in moderate use (i.e., completing 40–60%) of the intervention prior to disengaging [200].

Exposure to intervention content is of the upmost importance to ensure therapeutic gains [201]. High rates of non-compliance also call into question the ecological validity of clinical trials examining digital interventions, suggesting that engagement with digital mental health interventions may differ substantially in real-world settings [200]. Research into intervention and participant-related predictors of study attrition is a commonly adopted approach to better understand the factors associated with engagement and disengagement. Interestingly, several reviews have failed to identify any baseline participant characteristics that reliably predict attrition rates [152,202] from digital therapies. Rather, there is evidence to suggest that the degree of human feedback accompanying the intervention as well as the sophistication of the digital platform employed are associated with higher program usage [152,198,201].

### 5.1. Human Support

On average, dropout rates reported in trials of Internet-based programs are substantially higher than those observed in face-to-face psychological treatment, which range from 18% to 26% [164,203,204]. Researchers often attribute this discrepancy to the minimal clinician contact included in online interventions as compared to traditional therapist-led CBT protocols [204]. There is some evidence to support this hypothesis. As part of feedback collected in ter Huurne et al.’s (2015) large RCT of an Internet-based ED self-help intervention, 21% of participants who dropped out of the trial cited that the lack of personal contact in the online delivery of the program contributed to their disengagement from the intervention [205]. Furthermore, a meta-analysis assessing computer-based psychological treatments for depression (*n* = 23) found that supported interventions were associated with significantly greater retention as compared to unsupported interventions. Specifically, the dropout rate for unsupported digital treatments was 74%, in comparison to 28% for therapist-supported treatments and 38% for administrative-supported trials [206]. Similarly, Linardon and Fuller-Tyszkiewicz’s (2020) meta-analytic review of smartphone-delivered interventions for mental health problems (*n* = 70) found that attrition rates were significantly lower in trials which included reminders/prompts for participants to engage with the treatment and also for trials that adopted an in-person or telephone study enrolment process with a researcher (as opposed to online enrolment) [152].

Despite preliminary evidence indicating a role for clinician support in improving retention, very little is known about the optimal intensity of support (e.g., frequency, nature) or ideal channel of delivery (e.g., synchronous, asynchronous, email, telephone, videoconference). As discussed in an earlier section, the RCT conducted by Aardoom et al. (2016) is, to our knowledge, the only existing trial of an ED digital intervention which aimed to directly compare the effectiveness of a clinician-supported and unsupported digital ED intervention [170]. In addition to observing no added value of clinician support in symptom reduction, they also reported no significant differences in dropout rates between the clinician-supported and unsupported arms. Additional research is necessary in order to gain a better understanding of ideal “dose” of clinician support that is needed to maximise both outcome and adherence. This is of the upmost importance in order to prevent inefficient use of resources to deliver clinician-supported intervention, if not evidenced to improve outcome over and above independent engagement with online treatment.

### 5.2. Digital Design

An additional factor thought to influence treatment engagement is the sophistication of the digital platform through which the intervention is delivered. When considering the broad range of content delivery platforms (e.g., written manual, CD-ROM, etc), findings from Beinter et al.’s (2014) meta-regression analysis of self-help interventions for BN and BED (*n* = 73) suggest that Internet-based interventions outperform the rest. The authors observed that the highest study dropout rates were reported for the older technology of CD-ROM interventions and the lowest rates were reported for Internet-based interventions [202]. Additionally, abstinence rates were highest for Internet-based interventions (38%), followed by bibliotherapy (31%), with CD-ROM interventions performing poorly (9%). These results indicate a role for the use of sophisticated, engaging technologies to address both adherence and outcomes. This is in accordance with the diffusion of innovations theory, which proposes that rich, interactive interventions are more likely to foster greater adherence through the use of engaging design and functionality (e.g., diverse multimedia formats, activities, quizzes, and self-assessments) that help the user to understand the content in a more personally relevant manner [207,208]. In order to maximise adherence to self-help treatments, it has been suggested that the personalised and interactive capacity of digital interventions be employed to mimic features of direct contact with a clinician, such as participant-specific feedback [209]. Unfortunately, however, the available evidence supporting the role of interactivity in digital ED interventions is scarce and concern has been raised regarding the limited use of personalised and interactive digital features in existing programs [166]. Further research is needed to understand the relationship between digital features of online ED treatment platforms and engagement of users.

## 6. Conclusions

BN is a psychiatric illness associated with significant impairment, including high rates of chronicity, psychiatric comorbidity, and mortality, as well as reduced quality of life and physical health issues [40]. In response to troubling figures regarding poor treatment accessibility for individuals with an ED, lengthy durations of illness prior to receipt of care, low rates of evidence -based treatment uptake and poor long-term outcomes, there has been growing interest in the use of digital platforms as a broadscale vehicle of delivery for psychological treatment in favour of costly, time-intensive, and in-demand therapist-led interventions [150]. While digital therapy is a relatively novel treatment modality, written self-help programs, which are derived from first-line, face-to-face, clinician-led treatment manuals for BN (CBT-BN), have a strong record of producing high-quality outcomes when delivered along with brief support from a clinician. However, book-based interventions are outdated in their interactive capacity and limited in their reach, while investment in digital modalities is increasingly rapidly. Existing research tells us that moderately sized reductions in key symptoms of BN can be achieved following the use of CBT-based, online self-help treatments. There are, however, some noteworthy limitations of the existing research, including a small number of controlled trials contributing to the reported effect sizes, as well as high dropout rates and poor adherence identified in most studies. Therefore, whilst it is known that the core components of CBT-BN are translatable and effective when delivered as a low-intensity treatment, what is less clear is an understanding of the elements required to maximise both outcome and completion rates of online self-help treatments. This requires investment into high-quality research, with an emphasis on the factors influencing efficacy, engagement, and completion rates. This includes the degree of clinician support, sophistication of the digital platform, and digital design features such as interactivity and personalisation. Digital interventions hold the potential to broaden the treatment landscape for individuals with BN. By better understanding the ways in which both access to and completion of digital interventions can be maximised, it is possible that resource-poor, stretched health systems can overcome key barriers to the receipt of care and successfully deliver effective treatment producing significant symptom reduction for the majority of the illness group.

## Figures and Tables

**Figure 1 ijerph-20-00119-f001:**
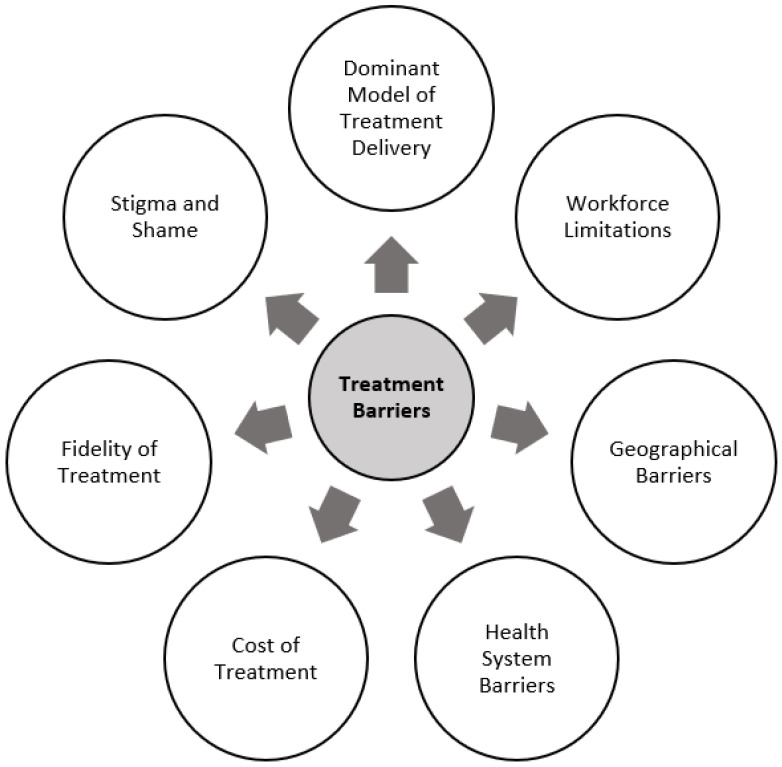
Barriers to treatment access for individuals with bulimia nervosa.

**Table 1 ijerph-20-00119-t001:** Diagnostic Criteria for Bulimia Nervosa derived from DSM-5-TR.

DSM-5-TR Diagnostic Criteria for Bulimia Nervosa
A. Recurrent binge-eating episodes, defined by: Eating a recognisably large amount of food within a discrete period of time; significantly larger than what the majority of individuals would consume in a similar situation and time frame; The episode is accompanied by a marked sense of a lack of control over one’s eating (e.g., feeling as though one is unable to stop eating or dictate the amount of food they consume). B. Repeated inappropriate compensatory behaviours with the aim of avoiding weight gain (e.g., self-induced vomiting, laxative misuse, diuretics, fasting, or exercise). C. On average, binge-eating episodes and compensatory behaviours are present at least once a week for 3 months. D. Self-evaluation is heavily influenced by concerns regarding body shape and weight. E. Such behaviours do not solely occur during episodes of anorexia nervosa.

**Table 2 ijerph-20-00119-t002:** Therapeutic strategies employed within Cognitive Behavioural Therapy for BN.

Phase	Content
One	- Psychoeducation - Personalised CBT-informed formulation regarding key maintaining processes - Behavioural techniques, including self-monitoring and regular eating
Two	- Cognitive techniques and behavioural experiments used to address avoidance behaviours and underlying psychopathology, including overvaluation of weight and shape
Three	- Relapse prevention, including strategies to maintain improvements and ways to address future setbacks

## Data Availability

Not applicable.

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
