# Peer review of "Accessibility of Psychological Treatments for Bulimia Nervosa: A Review of Efficacy and Engagement in Online Self-Help Treatments"

_ijerph, 2022, doi:10.3390/ijerph20010119_

Round 1

Reviewer 1 Report

The article is well written, the text is clear and easy to read, the results and data are well presented, and the main issue is clearly addressed, as well as what is relevant for the development of online self-help treatments, both as prevention and cure.  

The article is publishable, but I suggest: 

- Revising the wording of the results and conclusions to make them more informative, as well as including the following references

- Revision and inclusion of the following bibliographical references, all by Spanish authors, given that there is a very strong line of research in this area in Spain: 

> Elías-Zambrano, E.; Jiménez-Marín, G.; Galiano-Coronil, A.; Ravina-Ripoll, R. (2021). Children, Media and Food. A New Paradigm in Food Advertising, Social Marketing and Happiness Management. International Journal of Environmental Research and Public Health 18, 3858. 

> Fanjul-Peyró, Carlos; López-Font, Lorena; González-Oñate, Cristina (2019). "Los adolescentes y el culto al cuerpo: influencia de la publicidad e internet de la publicidad e internet en la búsqueda del cuerpo masculino idealizado". Doxa comunicación, n. 29, pp. 61-74. https://doi.org/10.31921/doxacom.n29a3

> Feijoo, Beatriz; López-Martínez, Adela; Núñez-Gómez, Patricia (2022). "The body and diet as sales arguments: Spanish teenagers' perception of influencers' impact on ideal physical appearance". Profesional de la información,v. 31, n. 4, e310412. https://doi.org/10.3145/epi.2022.jul.12.

> Jiménez-Marín, G.; Bellido-Pérez, E.; Trujillo Sánchez, M. (2021), Publicidad en Instagram y riesgos para la salud pública: el influencer como prescriptor de medicamentos. Revista Española de Comunicación en Salud 12 (1), 43-57.

Author Response

We thank Reviewer 1 for their comments and opportunity to improve the manuscript. We are appreciative of the reviewers' efforts to alert us to similar research that has been conducted by our Spanish colleagues, however upon reading each of the four papers suggested for inclusion we do not believe that they are relevant to the current literature review. The research focus of each of these papers relates solely to the impact of social media and Internet advertising upon body image and eating behaviours. Whilst this is a very important area of research, it is markedly distinct from the objective of the current literature review which is to summarise the way in which the Internet can be used to deliver evidence-based interventions, as opposed to it's harmful effects. We therefore do not think it is appropriate to include these references in the literature review, however are open to clarification from the reviewer regarding their perception of the link between our literature review of Internet-delivered treatment-focused research and the research findings presented in the references that they have suggested. 

Reviewer 2 Report

This is a very well written review of bulimia nervosa which could be interested by many researchers in the same field. The article could be improved by including few figures to summarize the symptoms of bulimia nervosa, corresponding treatment, and treatment barriers. These figures may help audiences to follow and understand. I would recommend this article for publish.

Author Response

We thank Reviewer 2 for their time taken to review our manuscript. We appreciate their positive feedback and opportunity to enhance the manuscript. We agree that inclusion of tables and/or figures will assist with readers' ease of understanding the material. Accordingly, we have included Table 1 which provides a summary the symptoms of bulimia nervosa as well as Figure 1 which outlines the barriers to treatment access. We are confident that these additions will address the reviewers' suggestion. 

Reviewer 3 Report

Thank you for the opportunity to review this manuscript.

Please see the attached document, which contains my review.

Author Response

We thank Reviewer 3 for their comprehensive feedback. Please find their comments addressed below. 

  • Reviewer's comment: Title: While it mentions some findings on the efficacy of digital interventions for ED and BN, the manuscript does not seem to review much on these treatments’ accessibility. Therefore, I hope the authors could reassess the title, perhaps one that reflects the main content of the manuscript. Since there is a designated section on engagement/adherence, I suggest the authors to consider this as part of the title.

Our response: We have revised and edited the title in accordance with the reviewer's comments: Accessibility of psychological treatments for bulimia nervosa: A review of efficacy and engagement in online self-help treatments. We believe that the revised title more accurately reflects the main content of the manuscript.  

  • Reviewer's comment: Abstract: A brief description of the methodology adopted by the authors would strengthen the abstract.

Our response: We have added the following sentence to the abstract: "A focused literature review was conducted, involving synthesis of a knowledgeable selection of high-quality, seminal articles, in order to provide an update on the current state of research in the field." We have adopted terminology of "a focused literature review" from Huelin et al. (2015) to help describe the methodology adopted in this paper. 

  • Reviewer's comment: Introduction/background: I must congratulate the authors for an interesting introduction. The literature review part is also very comprehensive. However, overall, Section 4 i.e. Digital Mental Health Interventions, is quite long and overinclusive. In my opinion, this section should focus immediately on the digital interventions for ED (briefly) and subsequently digital interventions for BN (as per title).

Our response: In accordance with both Reviewer 3 and Reviewer 4's comments we have significantly shortened Section 4 by removing subsections on COVID-19 (Section 4.1) and Digital Interventions for Non-Eating Disorder Mental Health Conditions (Section 4.2). We believe that by doing so this has addressed the Reviewers' recommendation to focus immediately on the research relevant to EDs, then BN. 

  • Reviewer's comment: Page 6 Line 262: “as they were not considered to be severely”.

Our response: This typo has been corrected. 

  • Reviewer's comment: Page 7 Section 2.6 Line 285: Can the authors elaborate a little bit more on this ‘therapist drift’?

Our response: We have included the following sentence to provide further detail regarding therapist drift: "Therapist drift occurs when a clinician deviates from evidence-based treatment protocols and is thought to contribute to the discrepancy between the outcomes achieved in clinical research trials and routine clinical practice (Waller & Turner, 2016)."

  • Reviewer's comment: Methods: Including the authors’ search method, and the inclusion and exclusion criteria (if any) would be good. This helps the reader understand the authors’ points, and whether review’s objectives and research questions were fulfilled by the search methods.

Our response: We have provided the following detail regarding methodology on page 2 of the revised manuscript: "The methodology adopted as part of the current review is consistent with that of a focused literature review (Huelin et al., 2015). Accordingly, we aim to present evidence from a knowledgeable selection of relevant, high-quality articles, including both individual trials of psychological interventions for bulimia nervosa (both therapist-led and self-help) as well as previous systematic reviews and meta-analyses. Given the review aims to provide a broad-reaching overview of the current state of research on self-help interventions for bulimia nervosa, no eligibility criteria were applied for included studies."

References

Huelin, R., Iheanacho, I., Payne, K., & Sandman, K. (2015). What’s in a name? Systematic and non-systematic literature reviews, and why the distinction matters. The Evidence, 34-37.

Waller, G., & Turner, H. (2016). Therapist drift redux: Why well-meaning clinicians fail to deliver evidence-based therapy, and how to get back on track. Behaviour research and therapy, 77, 129-137

Reviewer 4 Report

I found the article really interesting, and very scientifically and clinically relevant. Access to psychological treatment is a major current issue, not only for individuals who suffer from eating disorders. I particularly appreciate the effort to diversify interventions and to verify their effectiveness in this context. I have some more general comments about the article:

The DSM reference is missing in several places (e.g., line 47, 103, 399). 

In general, I find that there are too many abbreviations in the text and it becomes confusing. 

I question the relevance of two sections in the article. First, it appears that the section entitled "Digital Healthcare in COVID-19" (line 606) provides a lot of detail about the pandemic itself and lacks results for the BN. Is this section necessary?

Second, the section "Digital Interventions for Non-Eating Disorder Mental Health Conditions" (line 641) is not specific to the issue discussed in the paper.  It would be better to remove it, in my opinion, since the paper is already long. This could thus help to lighten it and focus more on your problem of interest. 

Here are also some more specific comments:

On line 195, I do not quite understand what you mean by "On average only 23.2% of individuals with – or at risk of – an ED access treatment".

On lines 230 to 232, what do you mean by "... which is not evidenced for their illness". 

In the section entitled "Health System Barriers" (line 254), it would be relevant to make a better link with the BN since this is the subject of your article. 

The section entitled "Fidelity of Treatment" (line 283) seems a bit short to me. Would you like to expand it? For example, you could make the connection to your subject of study, the BN?

In the "Stigma and Shame" section (line 291), the link to the BN should be made.

There is a reference missing in line 303, when you say : « It is well-established that the 301 receipt of care is compromised for individuals with BN. However, this stands in contrast to the range of psychological therapies that have been evaluated for the treatment of BN ».

On lines 306 to 310, could you better justify the choice of treatments that will be discussed? Also, it would have been relevant to see more statistics related to their effectiveness.

It seems to me that Table 1 (line 434) is not an addition to the text. Remove it?

At 527, when you say: "In addition to being an efficacious treatment option, self-help interventions accommodate for delivery of the effective components of treatment in a manner that places lower demands on resources". Can you qualify by saying that self-help can help to better understand and reduce some of the symptoms of BN while waiting for a more validated treatment?

Author Response

We thank Reviewer 4 for their time taken to provide detailed feedback on our manuscript. We have address their comments below. 

  • Reviewer comment: The DSM reference is missing in several places (e.g., line 47, 103, 399). 

Our response: We have reviewed the manuscript and added the DSM reference where missing. 

  • Reviewer comment: In general, I find that there are too many abbreviations in the text and it becomes confusing.

Our response: We have reviewed the manuscript and removed abbreviations on several instances. We have also included a list of abbreviations at the beginning of the manuscript to assist readers.

  • Reviewer comment: I question the relevance of two sections in the article. First, it appears that the section entitled "Digital Healthcare in COVID-19" (line 606) provides a lot of detail about the pandemic itself and lacks results for the BN. Is this section necessary? Second, the section "Digital Interventions for Non-Eating Disorder Mental Health Conditions" (line 641) is not specific to the issue discussed in the paper.  It would be better to remove it, in my opinion, since the paper is already long. This could thus help to lighten it and focus more on your problem of interest.

Our response: As suggested by the reviewer we have removed both sections entitled "Digital Healthcare in COVID-19" and "Digital Interventions for Non-Eating Disorder Mental Health Conditions". 

  • Reviewer comment: On line 195, I do not quite understand what you mean by "On average only 23.2% of individuals with – or at risk of – an ED access treatment".

Our response: We have edited this sentence to provide further clarity: "On average only 23.2% of individuals with a diagnosable ED seek treatment."

  • Reviewer comment: On lines 230 to 232, what do you mean by "... which is not evidenced for their illness".

Our response: We have edited this sentence to provide further clarity: "... which is not in accordance with evidence-based treatments for their illness".

  • Reviewer comment: In the section entitled "Health System Barriers" (line 254), it would be relevant to make a better link with the BN since this is the subject of your article.

Our response: Much of research into health system barriers has been conducted regarding EDs as a broad diagnostic category, rather than specific sub-categories of EDs. Therefore, we are unable to provide more BN-specific research on this matter as unfortunately the research is simply not available. 

  • Reviewer comment: The section entitled "Fidelity of Treatment" (line 283) seems a bit short to me. Would you like to expand it? For example, you could make the connection to your subject of study, the BN?

Our response: We have added to the "Fidelity of Treatment" section by providing a more in-depth description of therapist drift. Similar to the previous comment, it is difficult to speak specifically to BN given that most research on therapist drift report on EDs more broadly with research specifically on BN lacking. 

  • Reviewer comment: In the "Stigma and Shame" section (line 291), the link to the BN should be made.

Our response: We have address this comment by providing further detail regarding the shame and embarrassment associated with binge eating behaviours of individuals with BN.